

# An empirical approach and practical framework for a decentralized Ethereum Ecosystem Index (EEI)

Manoel Fernando Alonso Gadi[1], Maximilian Schmidt[2], Noah Ruemmele[2] and Miguel-Angel Sicilia[1]

[1] Department of Computer Science, University of Alcalá de Henares, Alcalá de Henares, Madrid, Spain
[2] Department of Computer Science, IE University, Madrid, Spain

## ABSTRACT

Stock market indices are pivotal tools for establishing market benchmarks, enabling investors to navigate risk and volatility while capitalizing on the stock market's prospects through index funds. For participants in decentralized finance (DeFi), the formulation of a token index emerges as a vital resource. Nevertheless, this endeavor is complex, encompassing challenges such as transaction fees and the variable availability of tokens, attributed to their brief history or limited liquidity. This research introduces an index tailored for the Ethereum ecosystem, the leading smart contract platform, and conducts a comparative analysis of capitalization-weighted (CW) and equal-weighted (EW) index performances. The article delineates exhaustive criteria for token eligibility, intending to serve as a comprehensive guide for fellow researchers. The results indicate a consistent superior performance of CW indices over EW indices in terms of return and risk metrics, with a 30-constituent CW index outshining its counterparts with varied constituent numbers. The recommended CW30 index demonstrates substantial advantages in comparison to established benchmarks, including prominent indices like DeFi Pulse Index (DPI) and CRypto IndeX (CRIX). Additionally, the article explores the practicality of implementing the CW30 in Layer 2 networks of the Ethereum Ecosystem, advocating for the Arbitrum infrastructure as the optimal choice for the decentralized crypto index protocol herein referred to as the Ethereum Ecosystem Index (EEI). The study's insights aspire to enrich the DeFi ecosystem, offering a nuanced understanding of network selection and a strategic framework for implementation. This research significantly enhances the existing literature on index construction and performance within the Ethereum ecosystem. To our knowledge, it represents a pioneering comprehensive analysis of an index that accurately mirrors the Ethereum market, advancing our comprehension of its intricacies and wider ramifications. Moreover, this study stands as one of the initial thorough examinations of index construction methodologies within the nascent asset class of crypto. The insights gleaned provide a pragmatic approach to index construction and introduce an index poised to serve as a benchmark for index products. In illuminating the unique facets of the Ethereum ecosystem, this research makes a substantial contribution to the current discourse on crypto, offering valuable perspectives for investors, market stakeholders, and the ongoing exploration of digital assets.

Corresponding author
Manoel Fernando Alonso Gadi,
manoel.gadi@uah.es

## INTRODUCTION

Stock market indices like the Standard & Poor's 500 (S&P 500), established in 1957, offer a strategic avenue for investors to diversify their portfolios. By gauging the performance of 500 leading U.S. companies, the S&P 500 provides a comprehensive view of the market's health. While the index itself is not directly investable, it can be accessed through specialized financial instruments. Investors can purchase exchange-traded funds (ETFs) or mutual funds that closely mirror the S&P 500's composition, thereby indirectly investing in it. This method of indirect investment in broad market indices serves as a risk-mitigation strategy, offering a level of diversification that can temper the impact of market volatility. Alternatively, investors can acquire shares in the companies that make up the S&P 500. However, to precisely recreate the index, this would cost a substantial sum of money and time. The development of an crypto index entails additional difficulties, ranging from the selection of eligible tokens with monetary value to the feasibility of implementing the index within an ecosystem. Such an index functions as a reliable reference, indicating the direction for investors who aspire to explore the novel territory of decentralized finance (DeFi) as well as the Ethereum ecosystem, enabling them to acquire exposure to a varied set of tokens while reducing the risks inherent in unregulated markets. However, constructing such an index is not a trivial task—it must surmount challenges such as variable transaction costs and the changing availability of tokens, some not desired like those with non-monetary value or fixed value in the case of stablecoins. Nevertheless, when constructed properly, an index be a remarkable vehicle and serve as a prominent financial product in the complex Ethereum ecosystem.

This article proposes an index for the Ethereum ecosystem, which has emerged as the leading smart contract platform, powered by its native token, Ether. Despite a downward trend in asset values, the number of smart contract deployments on Ethereum has continued to rise, indicating a robust and growing ecosystem.

The development of an crypto index entails additional difficulties, ranging from the selection of eligible tokens with monetary value to the feasibility of implementing the index within an ecosystem. This study aims to bridge this knowledge gap by constructing the Ethereum Ecosystem Index, which consists of tokens that represent ownership rights. This index functions as a reliable reference for investors who aspire to explore the novel territory of decentralized finance (DeFi) as well as the Ethereum ecosystem.

DeFi has experienced exponential growth, with novel trustless financial instruments gaining popularity. Index-tracking solutions have also found their place in DeFi, with Defi Pulse Index (DPI) emerging as a leading contender. Indices hold a prominent position in traditional finance and may be even more significant in DeFi due to the unprecedented degree of unregulated freedom that allows anyone to create a token.

The vast array of substantially different tokens presents a unique challenge for the average investor. A well-researched and reliable index solution provides a more secure way of gaining exposure to a diversified basket of tokens with inherent value. The research

process involves an extensive comparison of Ethereum mainnet and Layer 2 networks based on their transaction fees, as well as Layer 2 networks between each other based on total value locked and token availability as critical metrics, with the main of proposing the network that most feasible for the index implementation.

The establishment of a decentralized index could diminish the need for traditional financial intermediaries, who typically manage Exchange-Traded Funds (ETFs) associated with such indices. This decentralized approach offers a more direct and streamlined channel for investments, sidestepping the bureaucratic and operational bottlenecks often found in the conventional financial system.

The remainder of this article is organized as follows. First, "Literature Review" discusses a literature review on Index Methodologies and Blockchain. "Index Creation Methodology" describes our methodology for index creation and its results. "Decentralized Crypto Index Implementation-Discussion and Limitations" covers the implementation of the index protocol. Finally, in "Conclusion", we conclude with some notes on the availability of the *corpus* and acknowledgments.

# LITERATURE REVIEW

The advent of the blockchain and tokens opened new possibilities in which transactions are conducted. The exponential growth of this asset class has demonstrated the potential of tokens as an investment class and their ability to facilitate digital transactions without the need for federal regulatory intermediaries. However, in some cases, traditional financial constructs are still required for onboarding into the crypto economy or keeping custody of crypto assets. Furthermore, the lack of regulation presents a significant challenge in navigating the ecosystem. At present, there is an overwhelming amount of cryptocurrencies available for purchase, and this number is rapidly increasing. The creation of a crypto index is essential in reducing volatility and attracting risk-aware investors to the market. In this literature review, we will examine index creation in both stock market asset classes and the crypto market, and propose our own approach.

## Index construction methodologies

To date, the majority of indices are based on the CW as presented in *Bolognesi, Torluccio & Zuccheri (2013)*. Asset management firms predominantly use market CW indices because of the inherent operational advantages they offer. The index maintenance demands minimal effort as the constituents' weightings automatically adjust in response to stock price fluctuations. Moreover, fewer rebalancings are necessary leading to diminished transaction costs. The S&P 500, which uses the CW methodology, ranks as the most popular index (*Schnitzler, 2018*).

Despite the prevalence of CW indices, the article by *Bolognesi, Torluccio & Zuccheri (2013)* was the first to comprehensively compare the CW methodology with the Equal-Weighted (EW) approach. The authors focused on the constituents of the DJ Euro Stoxx index from January 2002 to December 2011 and provided empirical evidence of the higher risk-adjusted returns achieved by EW portfolios in comparison with CW indices. The authors demonstrated that the difference in performance between the two methodologies

could not be solely attributed to a 'size effect', which typically explains the variance in their outcomes. They confirmed their results by performing a Fama-French three-factor regression analysis. Bolognesi found that within the European equity market, the EW methodology exhibited higher excess returns in comparison to the CW approach. In addition to the popular CW and EW approaches, other construction methodologies have attracted interest in recent decades. These methodologies include fundamental-weighted (*Arnott, Hsu & Moore, 2005*), momentum-weighted (*Jegadeesh & Titman, 1993*), volatility-weighted (*Cazalet, Grison & Roncalli, 2013*), and multi-factor indices (*Amenc, Goltz & Lodh, 2012*). All approaches come with trade-offs, such as factor exposure, diversification, turnover, and tracking error. Therefore, market participants must carefully evaluate each methodology's objectives, assumptions, and implications before selecting an appropriate index.

## Existing crypto indexes

Over the years, various approaches have been employed to develop indices for the crypto market, often building upon established methodologies used in traditional market indices. One of the first significant attempts at creating a passive index for the crypto market was the development of the CCI30 in January 2017 (*Rivin & Scevola, 2018*). This index was designed as an investment vehicle for investors seeking to participate in the emerging crypto market. Due to the relative dominance of Ethereum and Bitcoin, as well as the extreme volatility of the market, the researchers employed an exponentially weighted moving average to calculate the market capitalization of tokens. This approach was chosen to avoid significant bias that would result from selecting a price on any given day. Additionally, only the top 30 coins by market capitalization were included in the index, as larger coins were generally subject to less volatility while still capturing 90% of the total industry market capitalization. The CCI30 index has been shown to have outperformed Bitcoin itself and has established itself as a reproducible index for passive funds and future ETFs.

One of the most notable research contributions in this area pertains to the Cryptocurrency Index (CRIX), an innovative index developed to reflect the overall market returns of cryptocurrencies. The origins of CRIX can be traced back to the seminal work of *Trimborn & Härdle (2018)*, who introduced the index in their article entitled "CRIX an Index for cryptocurrencies" with the aim of constructing a more comprehensive, stable, and diversified representation of the crypto market landscape. The methodology supporting CRIX relies on both the market capitalization and trading volume of cryptocurrencies and exhibits a dynamic nature. This dynamic approach is evident in the fluctuating number of index constituents that respond to market conditions. Researchers compute and maintain the index through a series of intricate steps that may initially resemble the methodology used in traditional market capitalization indices. However, the incorporation of a dynamic approach to ascertain the optimal number of tokens introduces a degree of complexity that far surpasses that of conventional indices.

The CRIX project was not alone in advancing crypto index studies. Building upon the foundation laid by the CRIX article, *Trimborn & Härdle (2018)* introduced a volatility

index known as VCRIX—A Volatility Index for cryptocurrencies. The primary objective of this index was to capture investor sentiment within the crypto market. VCRIX is similar to benchmarks such as VIX and VDAX, which measure volatility for popular equity indices such as S&P 500 and DAX, respectively. These indices are often referred to as "fear indices" due to their tendency to rise when investors are uncertain about future market direction. To construct VCRIX, the authors employed a heterogeneous autoregressive (HAR) model. As such, VCRIX shows potential as a useful indicator for capturing the complex dynamics of the rapidly evolving crypto market.

Prominent centralized exchanges have introduced index products to their offerings. Binance has introduced Binance CMC Cryptocurrency Top 10 Equal-Weighted Index. Bitpanda provides a diverse array of indices based on various asset classes, encompassing the top ten cryptocurrencies by market capitalization, smart contract platforms, decentralized finance leaders, and media and entertainment leaders. Crypto index products have also extended to neobanks and traditional brokers, with Germany's broker Comdirect being an example.

More experimental approaches have also been tested, one being Principal Component Analysis (PCA). *Shah, Chauhan & Chaudhury (2021)* proposed a mathematically robust dynamic index that avoids the intrinsic bias suffered by other rule-based crypto indexes. The researchers examined prices from 3 years' worth of data using PCA and found that dynamically updated indexes, meaning those updated periodically, performed the best. However, this approach had certain limitations that led the researchers to advise against its use as an index method. In particular, it captured the maximum volatility available, leading to a sub-optimal Sharpe Ratio, which calculates returns in relation to its corresponding volatility, resulting in a less favorable risk-return ratio. Another approach examines the factors influencing prices in the crypto industry. Sovbetov from the London School of Commerce analyzed six well-established cryptocurrencies in relation to an index of 50 cryptocurrencies he chose, known as the CRX50 (*Sovbetov, 2018*). By comparing the differences between these cryptocurrencies, he was able to determine factors that resulted in their differences in performance. In particular, Sovbetov found that price increases were related to several factors: Beta (prices increased when the Beta increased), trading volume (price increased with high weekly trading volumes), volatility (price decreased when the index was volatile), attractiveness (prices increased when online searches for the specific token increased), SPP (prices increased in the long-run when the SPP market value increased), and error correction terms (disequilibrium in crypto prices were corrected by 10–20% in the long-run).

For this article, one class of indices has captured our attention: smart contract-based indices. The most prominent example is the DeFi Pulse Index (DPI) (*DPI, 2023*), which exclusively targets investments in DeFi protocols. This innovative index represents a significant milestone, having amassed a maximum of over $200 million in assets under management by November 2021. Index Coop, the organization responsible for the index maintenance and deployment of the DPI smart contract, has also introduced alternative indices, including the Metaverse Index (*MVI, 2023*). An analysis conducted by DeFi Llama, a prominent analytics tool, identified over 40 decentralized protocols with a focus on either

index infrastructure or index curation. Moreover, social trading platforms, such as TokenSets (*TokenSets, 2023*), facilitate access to a plethora of indices. TokenSets allows organisations and individuals to create and launch bespoke indices, which other investors can then pursue and invest in. However, it is important to note that most of these indices have not achieved substantial assets under management.

## Research question

Following our review of the existing literature, we identified a need for a study that would develop an Ethereum Ecosystem Index (EEI), designed to track the market return of the Ethereum ecosystem and aiming to address the following research questions:

Research Question 1: What is the optimal methodology for constructing EEI in a manner that accurately reflects the Ethereum ecosystem, as determined through a comparative analysis of Capitalization-Weighted (CW) and Equal-Weighted (EW) index methodologies?

Research Question 2: Can EEI serve as an effective benchmark for evaluating the performance of investment strategies within the Ethereum ecosystem and the broader crypto community?

The first research question aims to identify the most effective methodology for constructing EEI by analyzing and comparing different index methodologies. The second research question seeks to determine whether the EEI can be utilized as a performance benchmark and as a foundation for potential index products.

## Significance of the study

This study makes several significant contributions to the existing body of knowledge on index construction and performance, particularly within the context of the Ethereum ecosystem. The significance of this research can be attributed to several key dimensions.

This study stands out as the first in-depth examination of an index meticulously crafted to encapsulate the dynamics of the Ethereum market, a domain that has attracted substantial interest in recent times. Ethereum holds a distinguished status as the foremost smart contract platform, providing the foundational layer for a multitude of protocols. Through its concentrated exploration of this market, the research enhances comprehension of Ethereum's intricate dynamics, shedding light on its far-reaching consequences for the crypto market at large.

This study is also among the first to conduct a thorough analysis of index construction methodologies within the emerging asset. By employing various analytical techniques to compare the performance of different index methodologies, this research provides a robust and nuanced understanding of the factors that drive index performance. Furthermore, the findings of this study offer a practical approach to index construction and propose an index that can be used as a benchmark for index products. By identifying the index methodology that yields superior performance in terms of financial indicators, this study has the potential to inform the development of successful index-based products with real-world applications.

In summary, by shedding light on the distinctive elements of the Ethereum ecosystem, this study makes a significant contribution to our current understanding of crypto and has broader implications for investors, market participants, and future research in the field of digital assets.

## INDEX CREATION METHODOLOGY

The following section describes the creation of the Ethereum Ecosystem Index (EEI) by comparing capitalization-weighted (CW) and equal-weighted (EW) index methodologies with varying numbers of constituents to accurately represent the Ethereum market and provide practical utility as a benchmark for investors.

Figure 1 the process of construction of the crypto index. The process commences with the manipulation of historical snapshots, which are supplied by CoinMarketCap. Following this initial step, the script evaluates the eligibility criteria for each token, a crucial phase in the index construction. Subsequently, price data is procured from a reputable source, Coingecko, and integrated into the methodology. The final stage involves the computation of the index, culminating in a comprehensive representation of the crypto market.

### Data

Reliance is placed on data from CoinMarketCap and Coingecko Pro API, widely recognized and trusted sources for obtaining accurate and reliable token price data. Leveraging these sources aims to provide a comprehensive overview of Ethereum prices and ensure the robustness of the studies. This approach enables effective back-testing of both CW and EW indices, ensuring a methodologically sound approach in terms of data availability and quality.

In order to uphold the principles of replicability and transparency, all pertinent data and codes available on the project's GitHub repository. This repository serves as a comprehensive resource for all information related to the project. Access the full repository here: (https://github.com/mkcschmidt/ethereum-ecosystem-index). Next, key insights about the data are presented.

The back-testing period for the index comprises 820 days and commences on the initial Sunday of 2021 (03/January/2021) and concludes on the final Sunday of Q1 2023 (02/April/2023).

Table 1 shows a five-row sample of the Aave token price and capitalization data retrieved from Coingecko. The sample is between December 31, 2022, and January 4, 2023. The table has three columns: 'timestamp', 'price', and 'market_cap'. The 'timestamp' column shows the date and time of the data point. The 'price' column shows the price of the Aave token in USD at that specific timestamp. The 'market_cap' column shows the market capitalization of the Aave token in USD at that specific timestamp. Market capitalization is calculated by multiplying the price of the token by its circulating supply. From this table, we can see that the price of the Aave token increased from $51.93 on December 31, 2022, to $56.83 on January 4, 2023, and its market capitalization also increased from $738,478,432.26 to $807,998,606.68 during the same period.

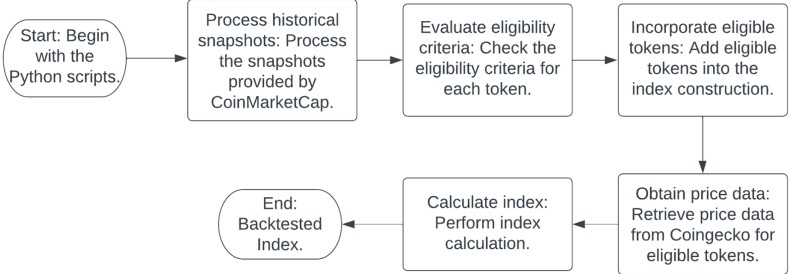

Figure 1 **The flowchart delineates the systematic procedure employed in the construction of the crypto index utilizing Python scripts.** This process commences with the manipulation of historical snapshots and culminates in the computation of the index. The methodology integrates data procured from credible sources and assesses the eligibility criteria pertinent to each token.

Table 1 **Five-row sample of the Aave token price and capitalization data retrieved from Coingecko.**

| Timestamp | Price | Market_cap |
|---|---|---|
| December 31, 2022 00:00:00 | 51.93 | 738,478,432.26 |
| January 1, 2023 00:00:00 | 51.99 | 739,535,655.86 |
| January 2, 2023 00:00:00 | 53.13 | 757,374,704.34 |
| January 3, 2023 00:00:00 | 53.05 | 754,392,167.03 |
| January 4, 2023 00:00:00 | 56.83 | 807,998,606.68 |

**Note:**
Sample between December 31, 2022, and January 4, 2023.

## *Data selection and limitation*

One of the fundamental decisions when creating an index is selecting the time period for analysis. The period between 2021 and 2023 was chosen for the creation of this crypto index for several reasons. Firstly, this period represents a time of significant growth and development in the crypto market, with the introduction of new technologies and the emergence of new players. Secondly, this period also saw increased regulatory attention and scrutiny, which had a significant impact on the market. Finally, by limiting the analysis to this specific period, it is possible to focus on recent trends and developments, while also acknowledging that the market is constantly evolving and that any analysis is subject to limitations. However, as a limitation of the current study, it should be noted that the crypto market has undergone significant changes during the current "bear market," which may distort the analysis.

## Token eligibility criteria

EEI serves as a benchmark reflecting the Ethereum ecosystem and employs rigorous inclusion criteria for selecting secure, valuable, and viable tokens. The criteria comprise four dimensions and are substantiated through the utilization of the International Token Classification (ITC) framework (*ITC, 2021*) developed by the International Token Standardization Association (ITSA) generously shared with us (ITC) allowing for efficient categorization of tokens based on established criteria.

The following outline the eligibility criteria for tokens included in the EEI, drawing inspiration from the International Token Classification (ITC) framework and incorporating relevant metrics for index classification to accurately represent the Ethereum ecosystem. Below, items 1 to 5 describe the token economic purpose, while items 5 to 7 describe the token's supply characteristics:

1. The token must be **part of the Ethereum ecosystem**.
2. The token must **not be classified as security** by the regulatory authorities in the respective jurisdictions of the United States and the European Union.
3. The token's Economic Purpose (ITC) dimension **must be one** of the following:

   - Utility Token: Governance Token
   - Utility Token: Ownership Token
   - Utility Token: Settlement and Governance Token

4. The token must be a bearer instrument, **excluding** the following Economic Purpose (ITC) dimensions from index inclusion:

   - Payment Token
   - Utility Token: Settlement Token
   - Utility Token: Access Token
   - Utility Token: Settlement and Access Token
   - Investment Token
   - Token with Other Economic Purpose

5. The token's supply must be reasonably **predictable** over a 5-year period, meaning that the token supply should not increase by more than 10% per year.
6. **Currently circulating at least 10%** of the anticipated 5-year supply.
7. The token's economics **must not entail locking, minting, or disadvantaging patterns** that would substantially disadvantage passive holders
8. The project must be widely **acknowledged** as developing a useful protocol or product.
9. Projects that primarily focus on competitive **trading or holding**, exhibit characteristics of a Ponzi scheme, or are predominantly designed for entertainment purposes are deemed ineligible for inclusion.
10. The project's protocol must exhibit **significant usage** exhibited by total value locked (TVL) (*Lacapra, 2022*) or user metrics.
11. The protocol or product must have been launched a **minimum of 180 days prior** to being eligible for inclusion in the index.
12. The protocol or project must **not be insolvent**.

Within the Economic Purpose dimension of the International Token Classification (ITC) framework, a critical criterion stands out (for an in-depth review), refer to the International Token Standardization Association document (ITC). The framework poses

the question: Does the token offer governance functionality related to the service, good, or functionality it grants access to, and/or to the overall environment? This specific criterion serves as a filtering mechanism for the EEI. It ensures the exclusion of multiple token categories such as payment and settlement tokens, wrapped tokens, tokenized derivatives, synthetic assets, tokens associated with physical assets, or tokens that represent claims on other tokens. To qualify, tokens must act as bearer instruments and embody some form of protocol ownership and governance participation.

EEI relies on the comprehensive set of inclusion criteria utilizing the International Token Classification (ITC) framework and data provided by the International Token Standardization Association (ITSA) to determine the tokens to be incorporated. This rigorous approach ensures the selection of secure, valuable, and well-supported tokens, providing a reliable and accurate benchmark for the Ethereum community and promoting greater transparency and safety within the global crypto market.

Table 2 presents Token name, Ticker, Coingecko ID, Key Person, USP, Economic Purpose, and Economic Purpose Description of a sample of five selected tokens sorter by rank and classified by the International Token Standardization Association (ITSA) using the International Token Classification (ITC) framework.

### Token eligibility—potential future limitation

One recent development that may serve as a potential limitation for this criterion is the classification made by the U.S. Securities and Exchange Commission (SEC). Although this classification has been taken into consideration thus far, it may soon render this criterion obsolete. In December 2020, the SEC filed a lawsuit against Ripple and its two top executives, Brad Garlinghouse and Chris Larsen, for selling XRP without registering it as a security. The SEC alleges that XRP buyers relied on Ripple's efforts to profit from the token, while Ripple contends that XRP is a payment tool rather than an investment. The trial could establish a precedent for crypto regulation in the United States. The outcome of the trial remains uncertain and could impact the price and status of XRP (*FOX Business, 2023*). The SEC has declared that at least 68 tokens are securities, affecting over $100 billion worth of tokens in the market (*Cointelegraph, 2023*). Tokens sold in initial coin offerings (ICOs) can be referred to by various names, but simply labeling a token as a "utility token" or structuring it to provide some utility does not preclude it from being classified as a security. According to the SEC, ICOs can present significant risks; while some may represent legitimate investment opportunities, many others may be fraudulent schemes designed to separate investors from their money with promises of guaranteed returns and future fortunes. They can also pose substantial risks of loss or manipulation, including through hacking, with little recourse available to victims after the fact (*U. S. Securities and Exchange Commission, 2023*).

Further regulations: The US Securities and Exchange Commission (SEC) has issued statements regarding the regulation of ICOs. The Howey test is utilized to determine whether a token qualifies as an investment contract, and thus a security (*Maume & Fromberger, 2019*). In the European Union, the definition of transferable securities under MiFID II and the draft proposal for a regulation on markets in crypto-assets (MiCA) aim

**Table 2 Sample of five selected tokens sorted by rank and classified by the international token standardization association (ITSA) using the international token classification (ITC) framework.**

| Token name | Concept | Value |
| --- | --- | --- |
| Amp | Ticker | AMP |
| Amp | Coingecko ID | Amp-token |
| Amp | Key person | Albert |
| Amp | USP | Provide assurance through collateralization for B2B focused crypto transactions. |
| Amp | Economic purpose | EEP22TU02 |
| Amp | Economic purpose description | Settlement and access token |
| The graph | Ticker | GRT |
| The graph | Coingecko ID | The-graph |
| The graph | Key person | Jan |
| The graph | USP | The Graph is an indexing protocol for querying networks like Ethereum and IPFS. |
| The graph | Economic purpose | EEP22TU03 |
| The graph | Economic purpose description | Settlement and governance token |
| Huobi token | Ticker | HT |
| Huobi token | Coingecko ID | Huobi-token |
| Huobi token | Key person | Marlene |
| Huobi token | USP | Native token of the cryptocurrency exchange Huobi Global. |
| Huobi token | Economic purpose | EEP22TU02 |
| Huobi token | Economic purpose description | Settlement and access token |
| Waves | Ticker | WAVES |
| Waves | Coingecko ID | waves |
| Waves | Key person | Aavneet |
| Waves | USP | Designed to enable users to create and launch custom crypto tokens. |
| Waves | Economic purpose | EEP22TU02 |
| Waves | Economic purpose description | Settlement and access token |
| TrueUSD | Ticker | TUSD |
| TrueUSD | Coingecko ID | True-usd |
| TrueUSD | Key person | Jan |
| TrueUSD | USP | The first regulated stablecoin fully backed by the US Dollar. |
| TrueUSD | Economic purpose | EEP21PP01USD |
| TrueUSD | Economic purpose description | USD-pegged payment token |

to standardize rules for crypto-assets that are not covered by existing EU legislation (*Hobza & Vondráčková, 2021*). For a comprehensive overview of the regulatory landscape for crypto-assets in various jurisdictions, including the US and the EU, please refer to the PwC Global Crypto Regulation Report 2023 (*PwC, 2023*).

## Capitalization-weighted (CW) and equally-weighted (EW) indexes

In the construction of EEI, capitalisation-weighted (CW) index methodology is employed as described in the article "A comparison between capitalization-weighted and equally weighted indexes in the European equity market" by *Bolognesi, Torluccio & Zuccheri*

*(2013)*. This study compares two major equity index construction methodologies: the capitalization-weighted and the equally weighted approaches. It focuses on the constituents of the DJ Euro Stoxx index from January 2002 to December 2011 and provides further evidence of the higher risk-adjusted returns achieved by equally weighted portfolios in comparison with cap-weighted indexes. The novelty of their study lies in testing these findings on the Euro stock market by using four reweighting frequencies (monthly, quarterly, semiannually, and annually) with the aim of identifying which is most able to maximize the benefits of the contrarian strategy implicit in the equally weighted approach.

A CW index, or value-weighted index, assigns weights to its constituents based on their market capitalization. This means that larger companies have a greater impact on the overall index. To study the effects of varying index constituents on the representation of the Ethereum market, three separate index versions will be constructed, containing 10, 20, and 30 constituents, respectively. The index will be reconstituted and rebalanced every 3 months to ensure it remains up-to-date. A normalizing divisor is used to represent the index price and is adjusted during each reconstitution and rebalancing phase. The index calculation is based on the Laspeyres index, which measures relative price changes over time using base period quantities such as market capitalizations.

The Laspeyres index is calculated as follows:

$$I_{n/0} = \frac{\sum P_n \cdot Q_0}{\sum P_0 \cdot Q_0}$$

The divisor plays a critical role in the calculation of the index value by ensuring consistency and continuity throughout reconstitution, rebalancing, and other market events. For example, if the index has a value of $500 and one constituent is replaced by another, the index must remain at $500 after the replacement. This continuity is achieved by making precise adjustments to the divisor. In the context of a Capitalization Weighted (CW) index, the initial divisor can be mathematically defined at the time of index creation as the initial total market capitalization of all constituents divided by the starting value that the index issuer aims to set when creating the index. Most commonly, the starting value of an index is chosen to be $100. When an adjustment to the divisor is required, the new divisor is calculated by multiplying the previous divisor by the ratio of the current total market capitalization to the previous total market capitalization.

The equal weight (EW) index calculation method assigns equal weights to each constituent asset. The equal weight for each constituent is found by dividing one by the total number of constituents in the index. The index level is calculated by summing the product of each token's price and its equal weight, representing the total adjusted value of all the index constituents, three separate index versions will be constructed, containing 10, 20, and 30 constituents, respectively. This method provides a balanced representation of the overall market, as it does not emphasize larger market capitalization but rather assigns equal importance to all constituents, regardless of their market capitalization. The divisor

**Table 3 Six constructed indices in this study with their methodology, number of constituents, and rebalancing interval.**

| Name | Index methodology | Number of constituents | Rebalancing interval |
|------|-------------------|------------------------|----------------------|
| CW10 | Capitalisation-weighted | 10 | 3 months |
| CW20 | Capitalisation-weighted | 20 | 3 months |
| CW30 | Capitalisation-weighted | 30 | 3 months |
| EW10 | Equal-weighted | 10 | 3 months |
| EW20 | Equal-weighted | 20 | 3 months |
| EW30 | Equal-weighted | 30 | 3 months |

is a constant value that normalizes the index level and must be adjusted accordingly to maintain index price continuity.

The procedure for divisor adjustment in the equal weight (EW) index mirrors the mathematical principles applied to the capitalization weighted (CW) index. The initial index value, set prior to index incorporation, serves as the basis for divisor calculation. Subsequent changes in the index, including reconstitution, rebalancing, and other events, necessitate corresponding adjustments to the divisor. These adjustments ensure the continuity and comparability of the index level over time. In the context of an EW index, the initial divisor can be mathematically defined at the time of index creation as the sum of the initial prices of all constituents divided by the number of constituents in the index and the specific starting value, usually $100. When an adjustment to the divisor is required, the new divisor is calculated by multiplying the previous divisor by the ratio of the current sum of prices to the previous sum of prices.

Table 3 summarizes the characteristics of six constructed indices in this study. The first (Name) lists the names of the indices: Capitalisation-weighted with 10 constituents (CW10), capitalisation-weighted with 20 constituents (CW20), capitalisation-weighted with 30 constituents (CW30), equal-weighted with 10 constituents (EW10), equal-weighted with 20 constituents (EW20), and equal-weighted with 30 constituents (EW30) The second column (Index Methodology) indicates the methodology used to construct each index. The indices with names beginning with "CW" are capitalization-weighted, while those beginning with "EW" are equal-weighted. The third column (Number of Constituents) shows the number of constituents in each index. The indices with names ending in "10" have 10 constituents, those ending in "20" have 20 constituents, and those ending in "30" have 30 constituents. The fourth column (Rebalancing Interval) indicates the frequency at which each index is rebalanced. All six indices are rebalanced every 3 months.

Figure 2 presents a graphical depiction of the performance of indices CW10, CW20, CW30, EW10, EW20, and EW30, over the period from January 3, 2021, to April 2, 2023. This encompasses their performance throughout the years 2021 and 2022, as well as the first quarter of 2023. By analyzing and comparing the performance of these indices as illustrated in Fig. 2, this study aims to ascertain the relative advantages and disadvantages of each index methodology. Furthermore, it seeks to identify the index that is most

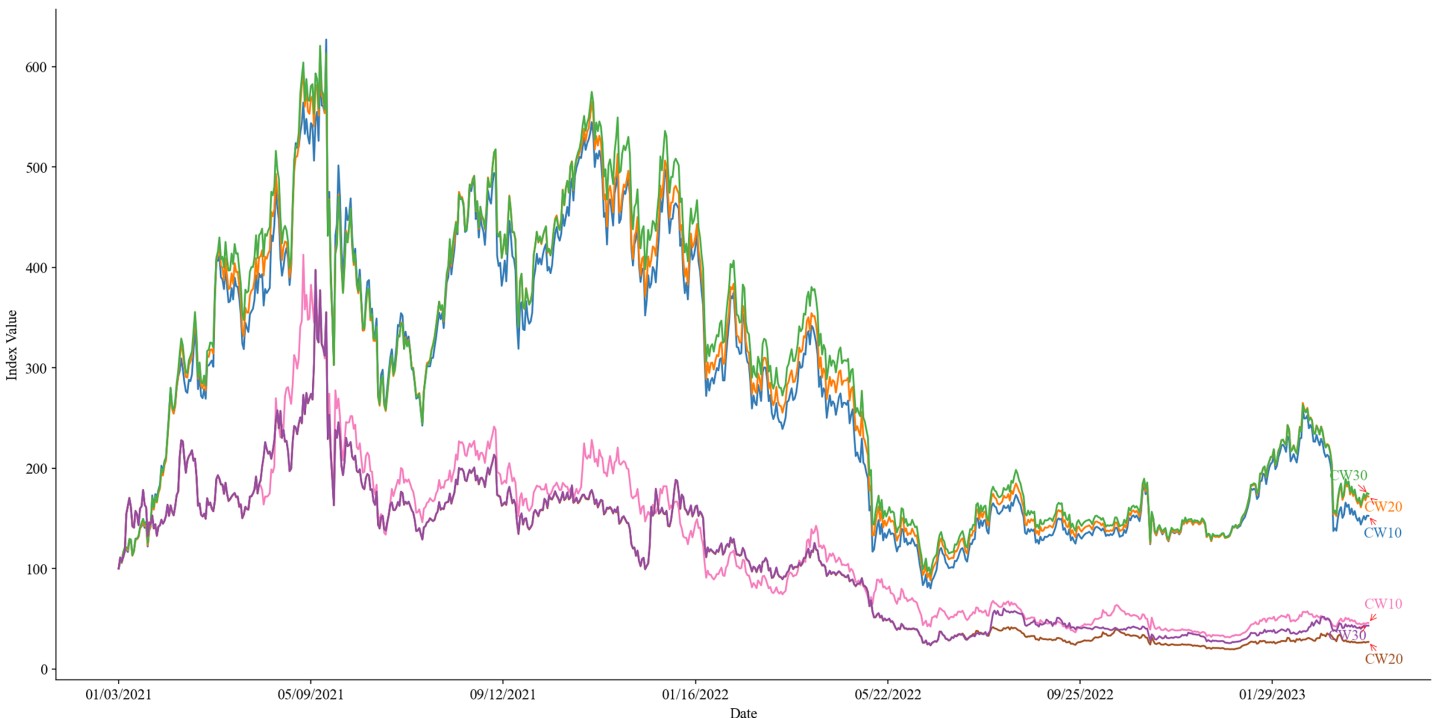

**Figure 2 Performance of indices CW10 (green), CW20 (yellow), CW30 (blue), EW10 (red), EW20 (purple), and EW30 (brown) from January 3, 2021, to April 2, 2023.**

**Table 4 Descriptive and performance statistics of the six created indices from January 3, 2021, to April 2, 2023.**

| Index | Total return | Mean return | Median return | Standard deviation | Sharpe ratio | Sortino ratio | Max drawdown |
|-------|-------------|-------------|---------------|--------------------|-----------|--------------|--------------|
| CW10 | 0.52461 | 0.00218 | 0.00491 | 0.05746 | −0.19770 | −0.29032 | 0.87228 |
| CW20 | 0.71665 | 0.00216 | 0.00458 | 0.05440 | −0.20827 | −0.30058 | 0.85645 |
| CW30 | 0.74345 | 0.00214 | 0.00544 | 0.05367 | −0.21101 | −0.30361 | 0.84857 |
| EW10 | −0.54273 | 0.00076 | −0.00097 | 0.05946 | −0.21459 | −0.36886 | 0.92396 |
| EW20 | −0.73165 | 0.00022 | 0.00083 | 0.06108 | −0.21789 | −0.35423 | 0.95102 |
| EW30 | −0.56969 | 0.00087 | 0.00081 | 0.06207 | −0.20530 | −0.32633 | 0.94078 |

**Note:**
The daily risk-free rate used in the calculation of the Sharpe and Sortino ratios is sourced from Yahoo Finance using the ticker symbol ^IRX.

effective in reflecting the Ethereum ecosystem and to provide valuable insights for investors and market participants.

## Empirical results

This section examines the performance metrics of three CW and three EW indices, constructed based on the number of constituents (10, 20, or 30) and a rebalancing period of 3 months.

Table 4 shows the descriptive and performance statistics of six indices from January 3, 2021, to April 2, 2023. The indices are divided into two groups: CW (cap-weighted) and EW (equal-weighted), with three versions of each based on the number of constituents (10,

20, or 30). The table displays various performance metrics for each index, including total return, mean return, median return, standard deviation, Sharpe ratio, Sortino ratio, and maximum drawdown. The CW indices generally outperform the EW indices in terms of total returns, mean returns, and risk measures such as standard deviation and maximum drawdown. However, both types of indices exhibit negative risk-adjusted performance as evidenced by their Sharpe and Sortino ratios. Among the six indices, the CW30 index demonstrates superior performance compared to the other indices.

Negative Sharpe and Sortino ratios, while typically indicative of unfavorable risk-adjusted returns, can sometimes be influenced by unique market conditions such as high volatility or extended bear markets. Therefore, selecting a different time frame could potentially yield more favorable risk-adjusted returns. While the risk-adjusted returns appear unfavorable within the chosen time frame, it is crucial to consider these metrics relative to the same time frame. Ethereum's exceptional total return does indeed set it apart among benchmark tokens. However, the index offers a superior Sharpe Ratio and significantly reduces risk through diversification. Lastly, the interpretation of these metrics heavily depends on individual investment goals and prevailing market conditions. An index that may seem unfavorable under certain metrics could still be the preferred choice for investors with specific objectives.

All indices generated (CW10, CW20, CW30, EW10, EW20, and EW30) failed to pass the normality test. As a result, Spearman's rank correlation was utilized in the analysis. This approach is consistent with the findings of Disci (2020), who demonstrated that Spearman's rank correlation is an effective tool for accurately ranking stock prices and identifying relationships between two variables. In their study, Disci (2020) compared gasoline prices in Turkey to Brent crude oil prices, and the results indicated that Spearman's rank correlation is highly effective in examining correlations between two variables.

Table 5 presents the results of a pairwise Spearman's rank correlation analysis between six indices: CW10, CW20, CW30, EW10, EW20, and EW30. The analysis was conducted over the time frame from January 3, 2021, to April 2, 2023. The table shows the correlation coefficients between each pair of indices, with the $p$-value for each correlation reported in parentheses. The correlation coefficients range from 0.848 to 0.998, indicating strong positive correlations between all pairs of indices. The $p$-values for all correlations are very small (most are close to zero), indicating that the correlations are statistically significant. The strongest correlations are observed between the pairs of indices within the same group (*i.e.*, CW10-CW20-CW30 and EW10-EW20-EW30), with correlation coefficients close to 1. The correlations between pairs of indices from different groups (*i.e.*, CW and EW) are also strong but slightly weaker than those within the same group. Overall, these results suggest that the six indices are highly correlated with each other over the time frame analyzed.

In conclusion, the correlation analysis provides valuable insights into the relationships between the CW and EW indices under various market conditions and segments. The analysis reveals the different exposures of these indices to various factors that affect their performance. While this analysis does not directly address the identification of the optimal

**Table 5 Spearman's rank correlation analysis between CW10, CW20, CW30, EW10, EW20, and EW30 from January 3, 2021 to April 2, 2023.**

|  | CW10 | CW20 | CW30 | EW10 | EW20 | EW30 |
|---|---|---|---|---|---|---|
| CW10 | 1 |  |  |  |  |  |
|  | (1) |  |  |  |  |  |
| CW20 | 0.995 | 1 |  |  |  |  |
|  | (0) | (1) |  |  |  |  |
| CW30 | 0.989 | 0.997 | 1 |  |  |  |
|  | (0) | (0) | (1) |  |  |  |
| EW10 | 0.844 | 0.842 | 0.835 | 1 |  |  |
|  | (1.7–208) | (1.7E−205) | (1.8E−199) | (1) |  |  |
| EW20 | 0.817 | 0.816 | 0.810 | 0.970 | 1 |  |
|  | (6.0E−183) | (1.6E−181) | (2.7E−176) | (0) | (1) |  |
| EW30 | 0.842 | 0.842 | 0.8352 | 0.944 | 0.977 | 1 |
|  | (2.3E−206) | (3.8E−207) | (1.4E−198) | (0) | (0) | (1) |

**Note:**
The *p*-value for each correlation is reported in parentheses.

index methodology, it provides important context for understanding the relationships between different index methodologies and their potential diversification benefits. Investors seeking greater diversification may need to consider other asset classes or the incorporation of alternative weighting methodologies. This supplementary analysis can be a useful tool for investors seeking a more comprehensive understanding of the Ethereum ecosystem.

Figure 3 presents a chart that illustrates the normalized performance of the CW30 index in comparison to prominent benchmarks, including Bitcoin, Ethereum, DPI, and CRIX, over the time frame from January 3, 2021, to April 2, 2023. The visualization highlights the performance of these indices throughout 2021, 2022, and the first quarter of 2023, demonstrating a strong relationship with Ethereum (commonly referred to as Ether).

Table 6 presents descriptive and performance statistics for the CW30 index and several benchmark tokens, including Bitcoin, Ethereum, DPI, and CRIX, over the time frame from January 3, 2021, to April 2, 2023. The table includes several key metrics for each index, including Total Return, Mean Return, Median Return, Standard Deviation, Sharpe Ratio, Sortino Ratio, and Maximum Drawdown. Total Return represents the overall return of the index over the specified time frame. Mean Return and Median Return represent the average and median daily returns of the index over the specified time frame. Standard Deviation measures the volatility of the index's returns. Sharpe Ratio and Sortino Ratio are measures of risk-adjusted performance, with higher values indicating better performance relative to the level of risk taken. Maximum Drawdown represents the largest peak-to-trough decline in the value of the index over the specified time frame. From the data presented in the table, we can see that Ethereum had the highest Total Return (1.34410) and Mean Return (0.00224) among all indices. CW30 had the highest Median Return (0.00544) and the lowest Maximum Drawdown (0.84857). Bitcoin had the lowest Total

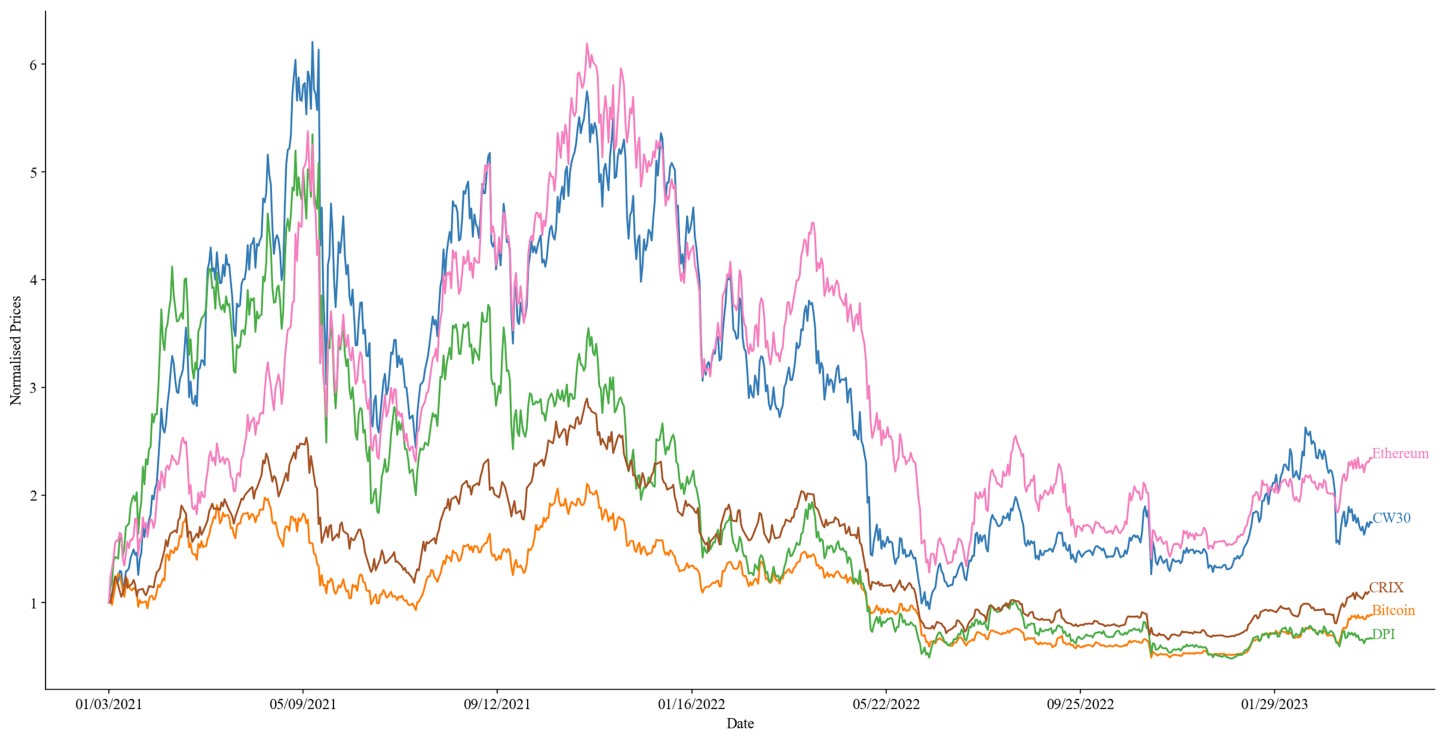

**Figure 3** Comparison of the normalized performance of the CW30 (blue) index against Bitcoin (yellow), Ethereum (purple), DPI (green), and CRIX (red), over the time frame from January 3, 2021, to April 2, 2023.

**Table 6 Descriptive and performance statistics for the CW30 index and benchmark tokens from January 3, 2021, to April 2, 2023.**

| Index | Total return | Mean return | Median return | Standard deviation | Sharpe ratio | Sortino ratio | Max drawdown |
|---|---|---|---|---|---|---|---|
| CW30 | 0.74345 | 0.00214 | 0.00544 | 0.05367 | −0.21101 | −0.30361 | 0.84857 |
| Bitcoin | −0.11394 | 0.00054 | −0.00027 | 0.03704 | −0.33257 | −0.51275 | 0.76718 |
| Ethereum | 1.34410 | 0.00224 | 0.00171 | 0.04882 | −0.22762 | −0.35232 | 0.79330 |
| DPI | −0.33342 | 0.00111 | 0.00119 | 0.05638 | −0.21934 | −0.33562 | 0.91005 |
| CRIX | 0.10029 | 0.00115 | 0.00141 | 0.04424 | −0.27288 | −0.41790 | 0.77302 |

Note:
The daily risk-free rate used in the calculation of the Sharpe and Sortino ratios is sourced from Yahoo Finance using the ticker symbol ^IRX.

Return (−0.11394) and Sharpe Ratio (−0.33257), while DPI had the lowest Sortino Ratio (−0.33562).

In conclusion, the CW30 index exhibits significantly better performance when compared to Bitcoin, DPI, and CRIX, particularly with respect to return metrics. The CW30 index demonstrates superior financial gains when compared to these benchmarks. However, the CW30 index shows partial underperformance relative to Ethereum over certain periods. Based on the performance metrics, the CW30 index may be considered a superior benchmark and may serve as a solid index for tracking the Ethereum market. Nevertheless, the comparison may have limited meaningfulness due to variations in the nature of the assets and the specific focus of the CRIX index. Further investigation is

**Table 7 Spearman's rank correlation analysis between CW30 and benchmark tokens (Bitcoin, Ethereum, DPI, and CRIX) from January 3, 2021, to April 2, 2023.**

| Token | Value |
|---|---|
| Bitcoin | 0.886 (4.3E−182) |
| Ethereum | 0.896 (3.0E−193) |
| DPI | 0.851 (2.5E−150) |
| CRIX | 0.926 (2.3E−233) |

**Note:**
The $p$-value for each correlation is reported in parentheses.

required to understand performance drivers and determine optimal investment strategies that take into account asset characteristics and investor objectives.

Table 7 presents the results of a Spearman's rank correlation analysis between the CW30 index and benchmark Bitcoin, Ethereum, DPI, and CRIX, over the time frame from January 3, 2021, to April 2, 2023. The Spearman's rank correlation measures the monotonic relationship between two variables, with values ranging from −1 (perfect negative correlation) to +1 (perfect positive correlation). The table shows the correlation coefficients between CW30 and each of the benchmark tokens, with the $p$-value for each correlation reported in parentheses. The $p$-value represents the probability of observing a correlation as strong as the one calculated if there was no true correlation between the variables. A small $p$-value (typically less than 0.05) indicates that the correlation is statistically significant. From the data presented in the table, we can see that CW30 has a strong positive correlation with all four benchmark tokens. The strongest correlation is observed with CRIX (0.926), followed by Bitcoin (0.886), Ethereum (0.896), and DPI (0.851). All correlations are statistically significant, with very small $p$-values close to zero. These results suggest that the price movements of CW30 are closely related to those of the benchmark tokens over the time frame analyzed.

It is important to note that the DeFi Pulse Index (DPI) concentrates on the Ethereum ecosystem. The CW30 index has demonstrated superior performance compared to DPI in several key metrics, including total return, mean return, and median return, while also exhibiting marginally lower volatility. Furthermore, the CW30's risk-adjusted performance, as measured by the Sharpe and Sortino ratios, is slightly superior to that of DPI. Additionally, the CW30 index has a lower maximum drawdown than DPI, indicating a smaller potential loss in a worst-case scenario. A notable distinction between the CW30 and DPI indices is the number of constituents. The CW30 index comprises 30 tokens, whereas DPI incorporates only 10. This difference in the number of constituents renders the CW30 index a more diversified investment option, potentially reducing portfolio risk while still providing exposure to the broader Ethereum market. In terms of approach, while DPI specifically focuses on DeFi protocols within the Ethereum ecosystem, the CW30 index incorporates all market segments, providing a more comprehensive representation of the entire Ethereum ecosystem. These distinctive traits of the CW30 index result in significantly better performance metrics compared to DPI.

## Comparison with related work

Our methodology presents several distinct advantages when compared to previous works. Here is a detailed comparison:

- Unlike the dynamic approach used by *Royalton (2020)* in the CRIX index to select and weight constituents based on their market size and liquidity, our method employs capitalization-weighted indexes with performance indicators that consistently outperform equal-weighted ones. While *Royalton (2020)* lacks a detailed explanation of the criteria for crypto inclusion in the index, our article offers a comprehensive analysis of various strategies, providing investors with transparency regarding both index composition and performance.

- While *Rivin & Scevola (2018)* proposed the CCI30 index, which includes only the top 30 coins by market capitalization, our method relies on market performance to determine index constituents. It involves a fixed pool of coins with weight adjustments based on market capitalization, employing a square root function, similar to our CW methodology. However, our approach stands out by providing a detailed comparison between various coins and EW/CW indices, offering investors valuable insights into performance disparities. Notably, *Rivin & Scevola (2018)* lacks a comparative analysis or empirical evidence regarding the performance of the CCI30 Index.

- In contrast to *Burggraf (2019)*, which used risk-based portfolio optimization strategies to construct crypto portfolios, our method introduces a user-friendly and inclusive approach accessible to all investors. Unlike the intricate strategies proposed in *Burggraf (2019)*, which may prove challenging for the majority of investors to replicate, our method provides a practical investment solution. This alternative to traditional benchmarks like the S&P 500 is designed to be straightforward and user-friendly, offering a complete package that can be readily utilized by investors.

- *Shah, Chauhan & Chaudhury (2021)* applied principal component analysis to construct and evaluate crypto indexes, finding that dynamically updated indexes performed better than static ones. Our method distinguishes itself by not only proposing an index designed to outperform the market, but also by providing investors with ownership through Ethereum smart contracts. This ensures that they have property rights over the assets and allows for performance assessment through market capitalization, especially during periods of market turbulence. In contrast to the PCA approach, which can prove unreliable during market fluctuations due to the difficulty in assessing correlations and predicting random market movements as discussed in *StatisticsGlobe (2023)*.

- *Chaudhari & Crane (2020)* analyzed the relationship between Bitcoin and other tokens, concluding that Bitcoin had a dominant influence on the crypto market. Our method, in contrast, offers a tangible investment solution, providing investors with a practical product and a wealth of comprehensive data. While *Chaudhari & Crane (2020)* primarily delves into Bitcoin's dominance and its interdependencies within the market, employing statistical measurements and metrics such as cross-correlation, eigenvalues/

vectors, and graph theory methodologies, our approach remains focused on delivering a concrete investment solution.

- Unlike _Sovbetov (2018)_, which examined factors influencing token prices for their weighting or selection criteria, our method uses a capitalization-weighted approach. This allows us to focus on market capitalizations rather than specific price-influencing factors.

- Finally, while _Disci (2020)_ demonstrated that Spearman's correlation is a valuable tool for ranking and discovering relationships between two variables, we use Spearman's correlation as a measure of similarity and comparison between the tested indexes and other existing indexes.

In summary, the current article offers an end-to-end solution for investors, providing transparent and user-friendly index options for tracking the Ethereum ecosystem's performance. Unlike some of the related works, our method aims to simplify the investment process, ensuring broad accessibility for investors while maintaining focus on strong index performance.

## DECENTRALIZED CRYPTO INDEX IMPLEMENTATION— DISCUSSION AND LIMITATIONS

The objective of the following section is to initiate a qualitative discussion regarding the most appropriate Ethereum network for the implementation of a crypto index protocol. The intention is to stimulate further research in this field and to demonstrate the feasibility of such an index. This section does not aim to provide a rigorous quantitative analysis or comparison. Instead, it seeks to foster a productive dialogue and encourage further exploration of this topic.

Table 8 shows an initial comparison between Layer 1 (L1) and Layer (L2) networks which suggests that deploying on L2 offers substantial cost savings in terms of transaction fees as token swap transaction fees on L2 are approximately 98% lower than on Ethereum.

The reduction of transaction fees on Layer 2 (L2) platforms is of particular importance for a crypto index protocol, given the involvement of multiple transactions during the rebalancing procedure. The significant decrease in fees on these platforms facilitates more straightforward processes on the protocol side and enhances accessibility and usability for investors. Based on the results of the L2 comparison, and pending further analysis, Arbitrum emerges as the initial target for deploying the protocol, followed by Optimism. The extensive and highly liquid token ecosystem of Arbitrum increases the potential for robust index curation, providing greater opportunities and exposure to a broader range of assets. Currently, both Optimism and Arbitrum offer a favorable balance between transaction costs and the number of tradeable tokens. The following paragraph offers an overview of the smart contract architecture that facilitates the basic index protocol required for EEI tokenization.

The infrastructure of EEI comprises several interconnected components, which can be assigned to either the core contracts, module contracts, or adapter contracts. The core contracts form the backbone of EEI as they define the primary logic required for the system

**Table 8 Gas fees single token swap Layer 1 (L1)/Layer 2 (L2) comparison.** The data for the economic comparison between deploying on L1 and the average L2 network was collected from various sources. For Arbitrum One, Optimism, Metis Network, Boba Network, and Polygon zkEVM, the data was obtained on April 16th, 2023 from L2Fees (L2Fees.Info, n.d.). For zkSync Era, the data was also collected on April 16th from the zkSync explorer. As there is no official source for average transaction fees on zkSync Era, two transactions were performed to serve as a proxy. For Arbitrum Nova and Ethereum, transaction cost data was collected on April 17th, 2023 from their respective network explorers. Transactional costs at that time were stable, and one transaction was chosen to serve as a proxy.

| Network | Gas fees for single token swap | Gas fees for 30 token swaps |
|---|---|---|
| Average layer 2 (L2) | $0.32 | $9.60 |
| Ethereum | $15.64 | $469.20 |

to function, managing critical components such as governance, token management, and system upgrades. A controller contract serves as the central registry for EEI, maintaining a list of approved modules, factories, and resources, and overseeing protocol fees and fee recipients. The integration registry allows for managing and tracking all external protocol integrations, holding a centralized registry of all integration adapters. The price oracle contract is responsible for obtaining asset pair prices from various sources like oracles or external adapters. The Index contract represents the index as a tokenized representation of its underlying constituents, including functionalities for managing the composition, applied modules, and characteristics of the index token. The Index Value contract determines the valuation of the Index Token using prices obtained from the price oracle contract.

The Index Value contract determines the valuation of the Index Token using price oracle data, accurately computing a single Index Token token's Net Asset Value. This valuation is vital for many functions of EEI, such as determining the required quantities of each constituent during minting and redeeming operations or evaluating the index's performance. Module contracts extend the functionalities of the core contracts, allowing developers to undertake smart contract iterations more flexibly and efficiently without affecting any core contracts. The basic issuance contract facilitates the minting and redeeming process of index tokens, while the index manager module functions as an automated mechanism for managing an index and allows for functionalities like periodic rebalancing periods. The streaming fee module is responsible for managing and accruing streaming fees for an index, and the trade module smart contract facilitates the management of index constituents through trade execution. These components work together to provide a robust and efficient infrastructure for EEI.

Table 9 compares various Layer 2 networks, focusing on the relationship between the networks' Total Value Locked (TVL) in escrow contracts on Ethereum as of April 14th, 2023, and the number of tradeable tokens within the set parameters through all available Decentralized Exchanges (DEXs) at the same time. The objective of this analysis is to draw a conclusion on which might be the most suitable L2 for deploying the protocol by assessing the validity of using TVL as a metric for concluding the number of tradeable tokens within the set parameters. The results show that Arbitrum has the highest TVL and

**Table 9 L2/L2 comparison.** TVL as of 14.04.2023 (total value locked in escrow contracts on Ethereum). Amount of tradeable tokens within the set parameters through all available Decentralized Exchanges (DEXs). EVM is a key piece of software that executes smart contracts on the Ethereum blockchain. TVL refers to the total value of assets locked within decentralized applications (dApps) or protocols on a blockchain.

| Name | Purpose | Technology | Ethereum virtual machine compatible (EMV) | Total value locked (TVL) | Amount of tradeable tokens |
|------|---------|-----------|--------------------------------------------|--------------------------|----------------------------|
| Arbitrum | Universal | Optimistic | YES | $6,860,000,000.00 | 172 |
| Optimism | Universal | Optimistic | YES | $2,130,000,000.00 | 59 |
| zkSync Era | Universal | ZK Rollup | YES | $243,000,000.00 | 16 |
| Metis Andromeda | Universal | Optimistic | YES | $133,000,000.00 | 16 |
| Arbitrum Nova | Universal | Optimistic | YES | $20,500,000.00 | 4 |
| Boba Network | Universal | Optimistic | YES | $13,980,000.00 | 4 |
| Polygon zkEVM | Universal | ZK Rollup | YES | $3,820,000.00 | 0 |

also the most tradeable tokens within the set parameters through all available DEXs. Furthermore, there is a positive correlation between the TVL and the number of tokens within the set parameters on the respective L2. This suggests that Arbitrum may be a suitable choice for deploying the protocol.

## Ethereum Layer 2 implementation disadvantages

Ethereum Layer 2 implementation involves trade-offs and limitations depending on their design and implementation (*Crypto University, 2021*), such as waiting periods for L1 withdrawals and security trade-offs. Furthermore, they currently lack the adoption and thereby liquidity of Ethereum L1. Moreover, newer L2s tend to be on the centralized side as there usually is a phase where the level of decentralization is kept low at inception for increased security in case of any network issues arising. Therefore a concern in the context of Layer 2 networks is the vulnerability of platform hacking. These attacks may target a specific Layer 2 protocol or a bridge that connects it to the main chain and may also come from inside of a particular project. The consequences of such attacks are systemic, as they may compromise the integrity and security of smart contracts or bridges, resulting in the theft of funds and the erosion of trust among users, platforms, and total value locked (TVL). Ethereum.org (*Buterin, 2014*) identifies several scenarios in which users' funds may be endangered, such as a bug in the smart contract code; a user error; a breach of the underlying blockchain; a malicious act by the bridge operators in a trusted bridge; a hack of the bridge itself. A recent example of such an incident is the Solana's Wormhole bridge hack, which resulted in the loss of 120,000 wETH ($325 million USD).

## CONCLUSION

This study presents a construction and analysis of six indices in the Ethereum ecosystem, focusing on CW and EW index methodologies with varying numbers of constituents. The research aims to identify the superior index methodology for EEI and subsequently propose an index that can accurately reflect the Ethereum ecosystem. The findings reveal

that the CW indices consistently outperform EW indices in terms of total returns, mean returns, and median returns. The CW30 index demonstrates the highest total return among the indices, indicating that larger CW indices may provide superior overall performance. Furthermore, CW indices exhibit lower risk, as evidenced by their lower standard deviation and maximum drawdown values. Overall, the analysis suggests that CW30 may provide an accurate representation of the Ethereum market and offer greater performance compared to EW indices. Additionally, the benchmark comparison evaluates the performance of CW30 against other prominent market indices and assets, including Bitcoin, Ethereum, DPI, and CRIX. The performance analysis demonstrated that CW30 outperformed Bitcoin, DPI, and CRIX in terms of total returns, mean returns, and median returns. This result supports the suitability of CW30 as a valuable benchmark and foundation for index products, catering to the needs of investors and other market participants in the Ethereum ecosystem and the broader crypto community. Finally, CW30 outperforms the current leading index, DPI, clearly showcasing its significant advantages and superiority.

The initial comparison for the most suitable Ethereum network for CW30 implementation suggests that deploying on L2 offers substantial cost savings in terms of transaction fees, which is particularly important for a crypto index protocol as multiple transactions are involved. The significant reduction in transaction fees on L2 platforms enables much easier processes on the protocol side and better accessibility and usability on the investor's side. Based on the L2 comparison results, Arbitrum emerges as the most obvious target for deploying the protocol, followed by Optimism. Arbitrum's extensive and highly liquid token ecosystem increases the potential for a solid index curation, offering better opportunities and exposure to a broader range of assets. At present, Arbitrum offers the best balance between transaction costs and the number of tradeable tokens. This suggests that Arbitrum may be a suitable choice for deploying the index protocol.

This study makes several significant contributions to the existing body of knowledge on index construction and performance, particularly within the context of the Ethereum ecosystem. The research is the first to our knowledge to conduct a comprehensive analysis of an index that reflects the Ethereum market and contributes to the understanding of the dynamics of Ethereum and its broader implications on the whole crypto industry. Additionally, this study is among the first to conduct a thorough analysis of index construction methodologies within the emerging asset class of crypto. The findings offer a practical approach to index construction and propose an index that can be used as a benchmark for index products. By shedding light on the distinctive elements of the Ethereum ecosystem, this study contributes significantly to the current understanding of tokens and has broader implications for investors, market participants, and future research in the field of digital assets.

In the future, there is potential for the newly proposed index, EEI, to be instantiated as a smart contract, culminating in the creation of a token compliant with the ERC-20 token standard (Ethereum Request for Comments 20). This compliance ensures the token's tradability across both centralized exchanges (CEXs) and decentralized exchanges (DEXs). The academic article provides a detailed framework for the deployment and maintenance

of the index, leveraging smart contract infrastructure. This innovative approach modernizes exchange-traded funds (ETFs) for the digital era, streamlining processes and potentially reducing investor fees by eliminating intermediaries. Furthermore, this methodology holds promise for extension to tangible assets, such as traditional securities, opening new avenues for asset tokenization and investment.

## Future work

Future work for this article could include a more detailed exploration of the security benefits of Ethereum over L2 scaling solutions. Additionally, the significance of decentralization and sufficient liquidity could be examined in greater depth. The role of regulatory bodies, such as the Securities Exchange Commission, in auditing and ensuring compliance could also be investigated.

Further work could involve experimenting with a wider range of constituents. Also, investigation into mitigating potential backtesting bias by choosing several backtesting periods could improve the results of the work. The plan also includes the integration of a mechanism for continuous assessment to enhance the index's relevance and adaptability, particularly in light of potential shifts in regulatory perspectives on token classifications. The importance of regular scrutiny and adaptation of the approach is acknowledged, ensuring compliance with legal boundaries. Notably, security eligibility criteria have already been established, addressing a significant portion of regulatory concerns.

Another potential direction for future work could be to expand the analysis to include additional time periods, both before and after the current bear market. This would allow for a more comprehensive understanding of the trends and developments in the crypto market and could help to mitigate the potential distortion caused by the current market conditions. Additionally, future work could also explore the impact of regulatory changes and other external factors on the market, as well as the emergence of new technologies and players.

It would be interesting to see researchers investigating the impact of the SEC's classification of tokens as securities on the crypto market and on the regulatory landscape.

Future research endeavors to refine and enhance the proposed index protocol may include investigations into cross-chain solution offerings, examinations of alternative mechanisms for maximizing the use of lockable constituents and increasing capital efficiency, implementing the protocol on the Arbitrum testnet to evaluate its functionality, and conducting practical tests of the EEI token to ensure it fulfills its established objectives. By addressing these future research directions, EEI can continue to evolve and adapt to the rapidly evolving DeFi ecosystem.

Finally, a compelling avenue for future research would be to explore the development of a secure Layer 2 solution or bridge for an index that effectively mitigates the risk of platform hacking and investigation on fluctuating transaction fees and token availability. This endeavor would entail conducting rigorous investigations into new security measures, examining the potential benefits of utilizing decentralized bridge operators, and undertaking further analysis of the Solana Wormhole bridge hack.

# ACKNOWLEDGEMENTS

We would like to express our gratitude to Joel Curado for sharing his feedback on the article.

### Funding
The authors received no funding for this work.

### Competing Interests
The authors declare that they have no competing interests.

### Author Contributions
- Manoel Fernando Alonso Gadi conceived and designed the experiments, analyzed the data, prepared figures and/or tables, authored or reviewed drafts of the article, and approved the final draft.
- Maximilian Schmidt performed the experiments, analyzed the data, performed the computation work, prepared figures and/or tables, and approved the final draft.
- Noah Ruemmele performed the experiments, analyzed the data, performed the computation work, prepared figures and/or tables, and approved the final draft.
- Miguel-Angel Sicilia analyzed the data, prepared figures and/or tables, authored or reviewed drafts of the article, and approved the final draft.

### Data Availability
The Python scripts for reproducing the generation and calculation of the crypto index based on Ethereum using the Coingecko Pro API are available at the Ethereum Ecosystem Index GitHub repository and at Zenodo:

- https://github.com/mkcschmidt/ethereum-ecosystem-index.

- Gadi, M. (2023). An empirical approach and practical framework for a decentralized Ethereum Ecosystem Index (EEI) datasets and codes [Data set]. In An empirical approach and practical framework for a decentralized Ethereum Ecosystem Index (EEI). Zenodo. https://doi.org/10.5281/zenodo.10251234.

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
