# Peer review of "An empirical approach and practical framework for a decentralized Ethereum Ecosystem Index (EEI)"

_PeerJ Computer Science, doi:10.7717/peerj-cs.1766_

## Round 0.1 · original submission · Major Revisions

The referral process is now complete. While finding your paper interesting and worthy of publication, the referees and I feel that more work could be done before the paper is published. My decision is, therefore to provisionally accept your paper subject to major revisions.

The referee comments are given below and on the website. Please make sure that you have addressed all the comments.

Reviewer 1 ·

Basic reporting

I have the following comments:
1.- The economic purpose descriptions, such as "Settlement and Access Token," could benefit further clarification. Providing concise examples or scenarios that illustrate each purpose would enhance understanding.
2.- The authors should enhance the clarity of the figures by providing distinctive labels and legends for each index line. The figures display lines in colors, but it's not explicitly indicated which index each line corresponds to. Adding readable and descriptive labels for each line and a legend associating colors with specific indices would significantly enhance reader comprehension, allowing users to quickly and accurately identify which index each line represents, enriching the visual experience, and facilitating the interpretation of the presented data.
3.- For enhanced verification, incorporating specific regulatory references or guidelines could support the requirement that tokens must not be classified as securities by regulatory authorities in the U.S. and the E.U. It could be helpful to include specific regulatory references or guidelines for easy verification.

Experimental design

I have the following questions:
1.-The criterion about the token's supply being "reasonably predictable over five years" might benefit from a more specific definition or guideline. What factors would be considered for supply predictability?
2.-Could the eligibility criteria be periodically reviewed and updated to address the potential impacts of evolving regulatory classifications, particularly in light of recent developments such as the U.S. Securities and Exchange Commission's actions against Ripple?
3.-How might incorporating a provision for ongoing assessment help the index maintain relevance and adaptability in the face of potential shifts in regulatory perspectives on token classifications?

Validity of the findings

The "CW30" index portray positive performance metrics like total and mean returns; However, do the negative Sharpe and Sortino ratios and the relatively high maximum drawdown signal that risk-adjusted returns were unfavorable? Furthermore, Ethereum's exceptional total return among the benchmark cryptocurrencies set it apart. Lastly, should interpretation consider the individual metrics and their interrelationships within the context of investment goals and market conditions?

Additional comments

The authors should split the research into two articles. In the first article, the authors should propose an index for the Ethereum ecosystem. In the second article, they should present the proposal for its implementation as a decentralized cryptocurrency index protocol and analyze its economic viability and practical performance within the DeFi ecosystem.

Lecwi: a capitalization-weighted index in the Ethereum ecosystem needs more clarity and provides a clear understanding of its content with further context.

·

Basic reporting

Here are some improvement suggestions for the authors of the research work:

Introduction Clarity: Clarify the introduction to provide a more concise and straightforward overview of the research problem, its significance, and the objectives of the study. This will help readers quickly grasp the context of your research.

Methodology Details: Expand on the methodology section to provide a more detailed explanation of how the index was constructed. Explain the criteria used for token eligibility and provide a step-by-step process for creating both capitalization-weighted and equal-weighted indexes.

Data and Analysis Transparency: Ensure that the data sources and analysis methods used are transparent and well-documented. Readers should be able to replicate your research if needed. Consider providing more information about the data sources for token prices and market capitalization.

Comparison with Related Work: Include a section that discusses how your research relates to and extends existing literature on cryptocurrency indexes. This will help establish the uniqueness and significance of your work.

Graphs and Visuals: Consider including graphs, charts, and visuals to illustrate key findings and trends in the data. Visual representations can make complex information more accessible to readers.

Consistent Terminology: Maintain consistent terminology throughout the paper. Ensure that abbreviations and acronyms are defined upon first use to avoid confusion.

Practical Implications: Discuss the practical implications of your research in more detail. How can investors and market participants use the proposed CW30 index, and what benefits can they expect? Address the real-world applications and implications of your findings.

Experimental design

Data Source Validation: Provide detailed information on the sources of cryptocurrency data used in your research. Ensure that the data is reliable, up-to-date, and comes from reputable exchanges or platforms. Consider using multiple data sources for robustness.

Methodology Transparency: Clearly document the specific formulas, algorithms, and parameters used in the calculation of the CW30 index. This will help readers understand how the index was constructed and enable them to replicate your methodology.

Sensitivity Analysis: Conduct sensitivity analysis to assess how changes in key parameters (e.g., constituent selection criteria, weighting methods) affect the performance of the CW30 index. This can provide insights into the robustness of your findings.

Validity of the findings

Validation Metrics: Expand the discussion on the metrics used to evaluate index performance. Include measures such as Sharpe ratio, Sortino ratio, and maximum drawdown to provide a comprehensive assessment of risk-adjusted returns.

---

## Round 0.2 · accepted · Accept

Since the reviewers are satisfied with the revised version, I am happy to inform you that your manuscript has been accepted for publication.

Reviewer 1 ·

Basic reporting

The authors improved the version of the article, considering all comments. In addition, they included new relevant references to support the proposal and provide more theoretical support.

Experimental design

The methodology and results were improved, allowing a better understanding of the author's contribution. In addition, the authors improved the visual presentation of the figures for a better understanding of the results.

Validity of the findings

The authors' contribution enhances the state of the art.

Additional comments

No comment

·

Basic reporting

Authors updated the paper.

Experimental design

As above

Validity of the findings

As above